# Functional Compartmentalization of HSP60-Survivin Interaction between Mitochondria and Cytosol in Cancer Cells

**DOI:** 10.3390/cells9010023

**Published:** 2019-12-19

**Authors:** Ya-Hui Huang, Chau-Ting Yeh

**Affiliations:** Liver Research Center, Chang Gung Memorial Hospital, Linkou, Taoyuan 333, Taiwan; e1249060@gmail.com

**Keywords:** heat shock protein 60, survivin, mitochondria, cytosol, apoptosis

## Abstract

Heat shock protein 60 (HSP60) and survivin reside in both the cytosolic and mitochondrial compartments under physiological conditions. They can form HSP60-survivin complexes through protein–protein interactions. Their expression levels in cancer tissues are positively correlated and higher expression of either protein is associated with poor clinical prognosis. The subcellular location of HSP60-survivin complex in either the cytosol or mitochondria is cell type-dependent, while the biological significance of HSP60-survivin interaction remains elusive. Current knowledge indicates that the function of HSP60 partly rests on where HSP60-survivin interaction takes place. HSP60 has a pro-survival function when binding to survivin in the mitochondria through interacting with other factors such as CCAR2 and p53. In response to cell death signals, mitochondrial survivin functions through preventing procaspase activation. Degradation of cytosolic survivin leads to the loss of mitochondrial membrane potential and aberrant mitosis processes. On the other hand, HSP60 release from mitochondria to cytosol upon death stimuli might exert a pro-death function, either through stabilizing Bax, enhancing procaspase-3 activation, or increasing protein ubiquitination. Combining the knowledge of mitochondrial HSP60-survivin complex function, cytosolic survivin degradation effect, and pro-death function upon mitochondria release of HSP60, a hypothetical scenario for HSP60-survivin shuttling upon death stimuli is proposed.

## 1. Introduction

The heat shock protein 60 (HSP60 or HSPD1), a mitochondrial chaperonin, is one of the abundant proteins in mitochondria of cells. Expression of HSP60 increases in various types of cancers, such as liver cancer, prostate cancer, and colorectal cancer, and higher HSP60 levels have been associated with tumor progression [1,2,3]. Clinical studies indicate that high HSP60 expression is associated with shorter survival in patients with serous ovarian cancer and gastric cancer [4,5]. Accordingly, HSP60 has been proposed as a biomarker for tumor growth and progression in several cancers, including breast [6], head and neck [7], colorectal [2], and pancreatic [8] cancers. HSP60 is capable of regulating apoptosis of tumor cells through interaction with several anti- or pro-apoptotic regulators such as Bax, Bak, p21, p53, and survivin [9,10,11]. The silencing of HSP60 results in inhibition of cell proliferation and promotion of apoptosis in tumor cells, suggesting its pro-survival and anti-apoptotic functions [8,12,13]. Nevertheless, the function of HSP60, in particular, the cytosolic HSP60, on the regulation of apoptosis appears to be more complex. The contradictory role of cytosolic HSP60 on the regulation of apoptosis has been demonstrated in different studies [14,15]. HSP60 in the cytosol has a pro-survival function, while it also participates in the apoptosis process upon cell death stimuli [15,16,17].

Survivin (BIRC5), the smallest member of the inhibitor of apoptosis protein (IAP) family, regulates cell division and inhibits apoptosis through blocking procaspase activation [18,19]. Among the members of the IAP family, survivin is the most highly expressed protein in fetal tissues and different types of human cancers. In contrast, survivin expresses at a very low level, if not totally absent, in healthy cells and tissues [20]. Owing to high expression of survivin in most human cancers, such as prostate, ovary, lung, breast, stomach and liver cancers, and the fact that silencing of survivin leading to cell proliferation inhibition and apoptosis of tumor cells, this protein has been proposed to be a candidate therapeutic target against cancer [21,22,23,24,25]. Similar to the distribution of HSP60 in cells, survivin also resides in both the cytosol and mitochondria of tumor cells. In both subcellular fractions, survivin has demonstrated a function of inhibiting apoptosis and promoting tumorigenesis [26,27]. Intriguingly, apart from the anti-apoptotic function, it has been argued that the cytosolic survivin also participates in the cell apoptosis process following death stimuli [26,28]. In this review, we take a closer look at this argument.

In tumor cells, HSP60 and survivin are capable of forming complexes to reside in either the mitochondria or the cytosol for survivin stabilization [9,12]. Consistent with the finding on stabilization of survivin by HSP60, silencing of HSP60 results in a decline of survivin expression [8,9,12]. As such, many similar biofunctions exist between HSP60 and survivin. In addition, a positive correlation has been demonstrated between their expression levels [12,29]. Nevertheless, our knowledge regarding the function of HSP60-survivin complex and the molecular events involved in tumor cells during apoptosis remains limited. Given this, the present review gathers reports on molecular functions of both HSP60 and survivin. Accordingly, a hypothetical model is proposed on how HSP60-survivin complexes participate in cell apoptosis upon death stimuli.

## 2. HSP60 Functions in Normal and Cancer Cells

Although HSP60 is abundant in the mitochondria, its expression can be found in both the cytosol and mitochondria under physiological conditions [30,31]. The role of HSP60 in apoptosis is quite conflicting. To support its anti-apoptotic function, knockdown of HSP60 has been found to promote apoptosis in cardiac myocytes, hepatocellular carcinoma (HCC), and breast cancer cells [11,12,32]. In an inducible cardiac-specific HSP60 knockout mouse model, it has been demonstrated that HSP60 deletion in adult hearts leads to an increased number of apoptotic cardiac myocytes, accompanied by altering mitochondrial complex activity, mitochondrial membrane potential (ΔΨm), and reactive oxidative species production. In addition, the HSP60 deletion is associated with altered levels of numerous mitochondria-localized proteins, including increased expression of pro-apoptotic Bax and decreased expression of anti-apoptotic Bcl-2. All these changes resulted in dilated cardiomyopathy, heart failure, and mortality [33]. In mouse B cells, HSP60 activates B cells and induces them to proliferate via toll-like receptor 4 (TLR4) signaling. Furthermore, HSP60 inhibits spontaneous or dexamethasone-induced apoptosis of B cells in a dose-dependent manner [34]. Contradictory to the pro-survival function, interaction between HSP60 and procaspase-3 has been demonstrated in HeLa and Jurkat T cells. Because HSP60 facilitates maturation of procaspase-3 in an ATP-dependent manner, it also harbors a pro-apoptotic function [35]. An HSP60 conditional transgenic mouse model has demonstrated that ubiquitous HSP60 induction in the embryonic stage leads to neonatal death in mice at postnatal day 1, wherein a high incidence of atrial septal defects have been observed with increased myocyte degeneration and apoptosis, leading to massive hemorrhage and formation of sponge-like cardiac muscles [36].

### 2.1. Pro-Survival Function of HSP60 in Mitochondria

The aforementioned anti-apoptotic or pro-apoptotic results are both obtained by altering the total amount of HSP60. However, because of the dual subcellular localization, the function of HSP60 as anti- or pro-apoptosis could be dependent on its location in cells and/or its ability to shuttle between mitochondria and cytosol. There are lines of evidence showing that mitochondrial HSP60 has a pro-survival/anti-apoptotic function because it interacts with or perturbes the functions of numerous survival regulators to participate in anti-apoptotic processes. For example, overexpressed mitochondrial HSP60/HSP10 is capable of suppressing doxorubicin-induced apoptosis in cardiomyocytes through increasing anti-apoptotic Bcl-x_L_ and Bcl-2, decreasing pro-apoptotic Bax, and inhibiting the activation of procaspase-3. In which, HSP60 not only interacts with Bax and Bcl-x_L_ but also inhibits the ubiquitination of Bcl-x_L_ [37]. In another study, by using high throughput proteomics screening, it has been shown that HSP60 stabilizes the mitochondrial pool of survivin. Acute ablation of HSP60 resulted in mitochondrial dysfunction and apoptosis. This process involves disruption of HSP60-p53 complex in mitochondria to stabilize p53, thus increasing Bax expression and Bax-dependent apoptosis [9]. Furthermore, in tumor cells, mitochondrial HSP60 acts as a regulator of mitochondrial permeability transition by association with cyclophilin D (CypD, a component of the mitochondrial permeability transition pore), which occurs in a multichaperone complex composed of HSP60, HSP90, and tumor necrosis factor receptor-associated protein-1 (TRAP1). HSP60 depletion induces CypD-dependent mitochondrial permeability transition, leading to apoptosis. Therefore, HSP60 functions as a cytoprotective chaperone antagonizing CypD-dependent cell apoptosis [38]. Taken to gather, all studies have consistently shown that HSP60 in mitochondria exerts a pro-survival/anti-apoptotic function albeit through various different molecular mechanisms.

### 2.2. Dual Functions of HSP60 in Cytosol

In contrast to the pro-survival/anti-apoptotic function of mitochondrial HSP60, the cytosolic HSP60 has been found to contribute to both anti-apoptosis and pro-apoptosis processes. In HeLa cells, cytosolic HSP60 participates in IKK/NF-κB activation by directly binding to IKKα/β. Loss of cytosolic HSP60 interrupts TNF-α-mediated activation of the IKK/NF-κB signaling, whereas cytosol-targeted expression of HSP60 promotes the IKK/NF-κB survival pathway, implying that cytosolic HSP60 has a pro-survival function [14]. In cardiac myocytes, cytosolic HSP60 blocks the pro-apoptotic ability of Bax and Bak to prevent apoptosis by forming the HSP60-Bax and HSP60-Bak complexes, also suggesting an anti-apoptotic function of cytosolic HSP60 [11]. In an animal model, elevation of cytosolic HSP60 and HSP70 expression has been observed in the neurons of the rostral ventrolateral medulla in Sprague–Dawley rats during pesticide mevinphos intoxication. Depletion of HSP60 or HSP70 enhances mevinphos mediated cardiovascular toxicity, with reduction of nitric-oxide synthase (NOS) I/protein kinase G signaling, enhanced NOS II/peroxynitrite cascade, and increased procaspase-3 activation [16].

Lines of evidence also suggest that HSP60 can contribute to pro-apoptotic functions. In a study conducted in Jurkat T cells, procaspase-3 forms a complex with HSP60 and HSP10 in mitochondria. Upon induction of apoptosis with staurosporine, HSPs are released from the mitochondria to the cytosol. Cytosolic HSP60 promotes the activation of procaspase-3 through cytochrome *c* and dATP in an ATP-dependent manner, suggesting a pro-apoptotic function [15]. Elevated sodium chloride concentration induces apoptosis in human umbilical vein endothelial cells and induces cytosolic and surface expression of HSP60 as well, which triggers autoimmune responses and leads to atherosclerosis [17]. However, in this case, the pro- or anti-apoptotic function of cytosolic HSP60 has not been clearly characterized. In a yeast system, it has been found that cytosolic HSP60 can stabilize Bax to enhance its association with mitochondria so that its pro-apoptotic effect is raised. In this model, it has also been found that cytosolic HSP60 physically interacts with proteasome to inhibit its activities, accompanied by the generation of an increased amount of polyubiquitinated proteins [39].

### 2.3. Mitochondrial HSP60 Shuttling to the Cytosol in Relationship to Its Functions

As stated above, HSP60 in mitochondria has a pro-survival/anti-apoptosis function, while in the cytosol, a dual function. When reviewing these studies, it appears that cytosolic HSP60 can either be derived from mitochondria or generated by de novo synthesis in the cytosol. There is an argument whether the pro- or anti-apoptotic function of cytosolic HSP60 is dependent on the origin of HSP60. In the study by Samali et al., it was found that increased cytosolic HSP60 abundance is due to its release from mitochondria to cytosol in staurosporine-induced apoptotic Jurkat T cells [15]. Similarly, in the study by Chan et al., it was also observed that a progressive decrease of mitochondrial HSP60 was accompanied by a gradual elevation of cytosolic HSP60 in rostral ventrolateral medulla neurons during mevinphos intoxication [16]. In contrast, the study by Chun et al. indicated that overexpression of cytosolic HSP60 (without affecting the mitochondrial level) enhanced the IKK/NFκB signaling pathway, which in turn promoted cell survival [14]. Chandra et al. have suggested that for increased cytosolic HSP60 to exert a pro-apoptotic function, it is necessary that mitochondrial HSP60 be released to the cytosol during apoptosis. Otherwise, de novo accumulation of cytosolic HSP60 would exert a pro-survival function [40]. Conceivably, this argument requires a detailed analysis of the participating cellular factors under different cellular conditions and models.

## 3. Survivin in Normal and Cancer Cells

In various normal tissues such as spleen and liver, survivin not only expresses at a low level but also resides exclusively in the cytosol [26]. However, in most of the tumor cells or tissues, survivin localizes both in the cytosol and mitochondria [12,26]. Recently, a study has discovered that survivin harbors a mitochondrial-targeting sequence located on its N-terminal region, suggesting that survivin can shuttle into the mitochondria from the cytosol dependent on the localization signal [41]. Intriguingly, similar to the functional compartmentalization of HSP60, mitochondrial survivin also has a pro-survival/anti-apoptotic function, whereas cytosolic survivin has an anti-apoptosis and an arguable pro-apoptosis function.

### 3.1. Pro-Survival Function of Survivin in Mitochondria

A study in 2004 by Dohi et al. firstly indicated the existence of mitochondrial survivin in cancer cells [26]. In 2016, Dunajová et al. have discovered that amino acids 1–10 at the N-terminus of survivin constitutes a mitochondrial-targeting signal, which is the critical sequence-motif responsible for survivin entering into the mitochondria [41]. In a study by Wheatly, survivin mutants comprising amino acids 1–120 and 11–142 were generated. Expression of both mutants was capable of protecting cells against TNF-related apoptosis-inducing ligand (TRAIL)-mediated apoptosis, whereas survivin 11–142 mutant could not protect cells from irradiation-mediated apoptosis [42]. These findings imply the pro-survival function of mitochondrial survivin. 

In the study by Dohi et al., it has been demonstrated that mitochondrial survivin exerts cytoprotective function through inhibition of apoptosis. They found that in HeLa cells, following hypoxia stimuli, the dramatic elevation of survivin levels was found in the mitochondria but only a minimal change was noted in the cytosol. Subsequent studies discovered that mitochondrial survivin was capable of inhibiting staurosporine-induced apoptosis by preventing the activation of procaspase-3 and procaspase-9. On the other hand, the tumor cells are more sensitive to cell death stimulation in the lack of mitochondrial survivin [26]. Notably, they also found that giving cell death stimuli to either breast cancer cells expressing cytosolic and mitochondrial survivin (MCF-7 cells) or insulinoma cells expressing only cytosolic survivin (INS-1 cells) results in different consequences. The stimuli did not lead to reduced viability in MCF-7 cells but resulted in the marked increase of cell apoptosis in INS-1 cells. Conversely, ectopic expression of mitochondrial survivin in INS-1 cells not only inhibited staurosporine-induced apoptosis but also promoted tumorigenesis. These findings help explain why mitochondrial survivin is only found in tumor cells but not in normal cells.

Besides exerting the anti-apoptosis function through inhibition of procaspase activation, mitochondrial survivin could also function via interaction with other mitochondrial proteins. For example, mitochondrial survivin can form a complex with HSP90 in mitochondria of vascular smooth muscle cells (VSMCs), stabilizing both proteins. In VSMCs suffering from shepherdin-induced apoptosis, both survivin and HSP90 were degraded. On the other hand, mitochondrial survivin levels increased in prototypic vascular growth factor (platelet-derived growth factor–BB)-induced VSMCs activation, suggesting a pro-survival function [43]. In another study, a mitochondrial protein, Smac/DIABLO, capable of release from mitochondria to the cytosol during apoptosis [44], was also found to interact with mitochondrial survivin [45,46]. During taxol-induced apoptosis, the formation of the survivin-Smac/DIABLO complex in mitochondria was essential for its anti-apoptosis function [45]. Disruption of the interaction between survivin and Smac in mitochondria facilitated the release of Smac from mitochondria to the cytosol and promoted apoptosis [46]. These findings unveil that mitochondria entry for survivin is critical for interaction and stabilization of specific mitochondrial proteins so as to overcome apoptosis in tumor cells upon death stimuli. Indeed, there are studies demonstrating that survivin accumulation in mitochondria enhances tumorigenesis through increased resistance to cell apoptosis in tumor cells [26,47].

### 3.2. Arguable Dual Function of Cytosolic Survivin

Not only the mitochondrial survivin exerts cytoprotective functions against apoptosis, but the cytosolic survivin is also capable of promoting cell survival. In a study by Fortugno et al., survivin has been found in both nucleus and cytoplasm. The cytosolic survivin is associated with interphase microtubules, centrosomes, spindle poles, and mitotic spindle microtubules during cell divisions and, therefore, possibly contributes a role in spindle assembly during mitosis [48]. Ectopic expression of survivin in the cytosol of 293T cells protects against etoposide-induced apoptosis [27]. Consistently, the ablation of cytosolic survivin leads to multiple centrosomal defects, abnormal spindle formation, and chromatin missegregation, which activate mitotic checkpoint by p53 induction. Complete ablation of cytosolic survivin further results in a marked reduction of ΔΨm, causing spontaneous apoptosis [49]. In the aforementioned study by Wheatly, in which survivin 1–120 and survivin 11–142 mutants were generated, it was found that survivin 11–142 mutant had no effect during mitosis, whereas survivin 1-120 prevented survivin from moving to the midzone microtubules during anaphase, suggesting a function of the C-terminus of survivin in the maintenance of normal mitosis.

Despite most studies have demonstrated that both mitochondrial and cytosolic survivin have an anti-apoptotic function, some studies suggest that the cytosolic survivin is also involved in pro-apoptosis processes of tumor cells following death stimuli, although the latter view is arguable. In the aforementioned study by Dohi et al., INS-1 cells stably expressing mitochondrial survivin (INS-1/MT-S cells) manifested survivin release from mitochondria and accumulation in the cytosol, accompanied by cell apoptosis upon death stimuli [26]. One could argue that the cytosolic survivin released from mitochondria contributes to apoptosis. Another noteworthy mechanism has been reported, involving the compartmental changes of the survivin-X-linked IAP (XIAP) complex. Under normal conditions, XIAP, an anti-apoptotic cofactor, binds to survivin in the mitochondria; however, the mitochondrial survivin-XIAP complex is released to the cytosol when tumor cells exposed to cell death stimuli. Subsequently, the binding interface of the cytosolic survivin-XIAP complex is disrupted owing to phosphorylation of survivin by cyclic AMP-dependent protein kinase A (PKA), resulting in XIAP degradation. Notably, cell death stimuli did not cause survivin phosphorylation and XIAP degradation in the mitochondria [47].

Different from the survivin-XIAP mediated mechanism, a later study has discovered that the XIAP is also a ubiquitin E3 ligase for survivin. Interferon-β-mediated XIAP-associated factor 1 (XAF1) induction promotes the formation of XIAP–XAF1–survivin complex to enhance survivin ubiquitination and degradation [28]. XAF1 is a tumor suppressor, which induces apoptosis and inhibits tumor growth in gastric cancer and HCC [50,51]. According to these findings, one could speculate that XAF1-induced apoptosis might result in survivin release from mitochondria to the cytosol to trigger the formation of XIAP–XAF1–survivin complex for further ubiquitination and degradation of survivin.

## 4. HSP60–Survivin Interaction

The interaction between HSP60 and survivin was first proposed in 2008 by Ghosh et al. They have demonstrated that binding of HSP60 to survivin occurs in the mitochondria but not in the cytosol of MCF-7 cells. The formation of mitochondrial HSP60–survivin complex assists the stabilization of mitochondrial survivin. Silencing HSP60 in MCF-7 cells triggers the activation of procaspase and results in cell death. This pro-apoptotic process can be prevented by overexpressing survivin simultaneously [9].

Similarly, our recent study also revealed the formation of HSP60-survivin complex in HCC cells (Huh7 and J7). However, different from what has been found in breast cancer cells, the HSP60–survivin complex in HCC cells only exists in the cytosol but not mitochondria. Nevertheless, the molecular stabilization effect of survivin through the formation of cytosolic HSP60–survivin complex in HCC cells is the same as what has occurred in breast cancer cells [12]. No matter where the HSP60–survivin interaction occurs, cytosol or mitochondria, the amounts of these two molecules are positively correlated [9,12,29,34]. Decreased expression of HSP60 leads to a decrease of survivin levels that eventually causes cell death [9,12]. Additionally, HSP60 expression is associated with increased levels of survivin in a dose-dependent manner [34].

### 4.1. Interaction between HSP60 and Survivin in Mitochondria

Both HSP60 and survivin in mitochondria play a pro-survival/anti-apoptotic function during cell death stimuli. The formation of HSP60-survivin complexes in mitochondria seems to be a critical event assisting cells to overcome death stimuli. For example, knockdown of mitochondrial HSP60 promotes apoptosis by disrupting HSP60–survivin interaction in tumor cells [40]. Recently, cell cycle and apoptosis regulator 2 (CCAR2) has been functionally linked to the HSP60–survivin complex to modulate cell death. CCAR2, a widely expressed protein, is required for cell proliferation and is associated with poor prognosis in squamous cell carcinoma [52]. CCAR2 silencing decreases cell proliferation by altering the AKT pathway [53]. Two studies have indicated that CCAR2 interacts with HSP60 and survivin to form a CCAR2–HSP60–survivin complex in mitochondria. CCAR2 appears to stabilize the formation of HSP60–survivin complex to protect the cells in neuroblastoma or breast cancer cells from mitochondrial stress-induced apoptosis. Additionally, CCAR2 binds to survivin in the absence of mitochondrial stress, and CCAR2–HSP60 complexes increase upon mitochondria stress. In this view, CCAR2 sequesters survivin and brings it to HSP60 in mitochondria, allowing HSP60 to stabilize survivin [29,54].

### 4.2. Interaction between HSP60 and Survivin in Cytosol

There is little dispute regarding the pro-survival/anti-apoptotic function of HSP60, survivin, and HSP60-survivin complexes in mitochondria. However, both HSP60 and survivin in the cytosol are involved in pro- and anti-apoptotic mechanisms, partly dependent on whether their cytosolic accumulation is due to their release from mitochondria. In tumor cells receiving death stimuli, it has been shown that HSP60 and survivin respectively accumulate in the cytosol, accompanied by decreasing amounts of them in mitochondria. The release of these proteins from mitochondria to cytosol partly contributes to death stimuli associated apoptosis [26,40]. In contrast, de novo or the extraneous introduction of these proteins into cytosol results in a cytoprotection function. To date, the release of HSP60-survivin complexes from mitochondria to the cytosol during apoptosis and the subsequent molecular events have not been clearly delineated yet. Based on the existing evidence, such a scenario is highly likely to occur. Whether in the tumor cells (such as breast cancer cells), where HSP60–survivin complex formed in the mitochondria, could manifest HSP60–survivin release into the cytosol during death stimuli and play a role in apoptosis awaits further investigation. 

On the other hand, it is important to investigate the molecular roles of HSP60–survivin complexes that exist only in the cytosol of tumor cells (such as HCC cells) during apoptosis. In many cancers, the expression levels between HSP60 and survivin show a positive correlation, and the levels of survivin are partly regulated by HSP60 [9,12,34]. In esophageal squamous cell carcinoma and HCC, in around 60% to 70% of cases, survivin is overexpressed in tumor tissues, while the percentages of over- and under-expression of HSP60 are largely equal [12,55,56]. The HSP60–survivin complex can stabilize survivin in the cytosol of HCC cells under physiological conditions [12]. As such, one can speculate that a relatively higher percentage of survivin overexpression is present in the tumor parts with a proportion of them stabilized by HSP60. Judging from these data, the overall effect of survivin should be pro-survival in cancer cells. The so-called “pro-apoptotic” mechanisms, such as the release of the mitochondrial survivin–XIAP complex to cytosol upon death stimuli, leading to the disruption of cytosolic survivin–XIAP complex and thus XIAP degradation [47], and induction of apoptosis by XAF1 through facilitation of cytosolic survivin degradation by forming XIAP–XAF1–survivin complexes [28,50,51] are in fact due to the “loss of pro-survival function” of mitochondrial survivin because of its release into cytosol. In this view, the efficacy of survivin degradation in cytosol might be a significant contributor to apoptosis. Notably, the reduction of cytosolic survivin causes the loss of ΔΨm, leading to spontaneous apoptosis [49]. The fact that the mitochondrial survivin is only found in tumor cells while cytosolic survivin exists both in normal and tumor cells implies that survivin degradation most likely occurs through a common pathway in the cytosol. This view is supported by the observation that survivin is released from mitochondria to cytosol upon death stimuli. In the study by Zhao et al., during the G1 phase, the degradation of cytosolic survivin in 293 cells is regulated by the ubiquitin-proteasome pathway [57]. In the study by Dohi et al, it was found that proteinase K only digests cytosolic but not mitochondrial survivin [26]. Therefore, survivin residing in the mitochondria of tumor cells has an additional benefit in that its degradation can be deterred, allowing for prolonged pro-survival/anti-apoptotic functions.

In the study by Kalderon et al., cytosolic HSP60 is capable of regulating proteasome activity in yeast. Elevation of cytosolic HSP60 induces the accumulation of polyubiquitinated proteins [39]. During death stimuli, increased cytosolic HSP60 due to mitochondrial release exerts a pro-apoptotic function in cells. The interplay between HSP60, survivin, and ubiquitination during apoptosis remains unclear. However, according to our understanding of HSP60 and survivin functions in tumor cells during apoptosis, a possible mechanism of HSP60–survivin complex participating in apoptosis of tumor cells upon death stimuli is proposed as shown in Figure 1. Without death stimuli, HSP60 and survivin form a complex (HSP60–survivin complex) for the stabilization of survivin. However, upon death stimuli, because of the change of cellular microenvironment, the function of HSP60–survivin can be altered. Route 1, the tumor cells harbor the existing HSP60–survivin complex in the mitochondria, where the HSP60–survivin complex is released from mitochondria to the cytosol during apoptosis. Route 2, the tumor cells harbor existing cytosolic HSP60–survivin complex as well as free survivin molecules in the cytosol, where the mitochondrial HSP60 is translocated to the cytosol to capture free survivin during death stimuli. Regardless of the routes (1 or 2), which lead to the increase of cytosolic HSP60–survivin complex, survivin will eventually undergo ubiquitination and degradation. A plausible hypothesis is that the formation of cytosolic HSP60–survivin complexes following death stimuli might somehow facilitate the ubiquitination of survivin for its degradation. The study by Kim et al. has demonstrated the formation of CCAR2–HSP60–survivin complex, while no detailed experiments were performed to explore the fates of the complex upon death stimuli. [29] A review by Kelly et al. suggested chaperone binding to survivin could form a stabilized complex. [58] Therefore, it remains possible that the heterocomplex could stabilize both proteins, whereas upon death stimuli, other factors might join in to destabilize them, leading to survivin degradation via ubiquitination.

## 5. Conclusions

There are many similar biological functions between HSP60 and survivin. Both of them are overexpressed in many cancer cells or tissues, especially for survivin. Silencing the total amount of HSP60 or survivin can inhibit cell proliferation and promote apoptosis in various types of tumor cells [9,12,24,38,59,60,61]. Survivin expression increases in cells during the G2/M phase of the cell cycle, followed by a rapid decline at the G1 phase [57]. Similarly, the accumulation of HSP60 is also observed at the G2/M phase in rat liver cells [62]. In tumor cells, they reside in the mitochondria and cytosol. The functions of HSP60 and survivin in mitochondria are both pro-survival and anti-apoptotic, albeit the molecular mechanisms are different. On the other hand, both HSP60 and survivin have been found to harbor dual function in the cytosol. They can participate either in the pro-survival or pro-death processes. Despite so many common functional features between HSP60 and survivin, researches rarely consider both proteins simultaneously when it comes to functional characterization. Moreover, HSP60 can physically interact with survivin to form HSP60–survivin complexes in various types of cancer cells. The formation of this complex stabilizes survivin and thus promotes cancer cell survival. After reviewing previous studies on HSP60 and survivin, we propose a hypothesis that depicts the possible molecular events occurring in HSP60–survivin complex following death stimuli in cancer cells. Under physiological conditions, cytosolic or mitochondrial survivin is stabilized by the formation of cytosolic or mitochondrial HSP60-survivin complex. Once the cells receive death stimuli, mitochondrial HSP60 or the HSP60–survivin complex is released to the cytosol, increasing the abundance of the cytosolic HSP60–survivin complex. Here we speculate that survivin will undergo degradation possibly through ubiquitination since the cells have proceeded to apoptosis. The detailed mechanisms and other players involved in these events have not been characterized, and the hypothesis remains to be tested and verified in the future.

## Figures and Tables

**Figure 1 cells-09-00023-f001:**
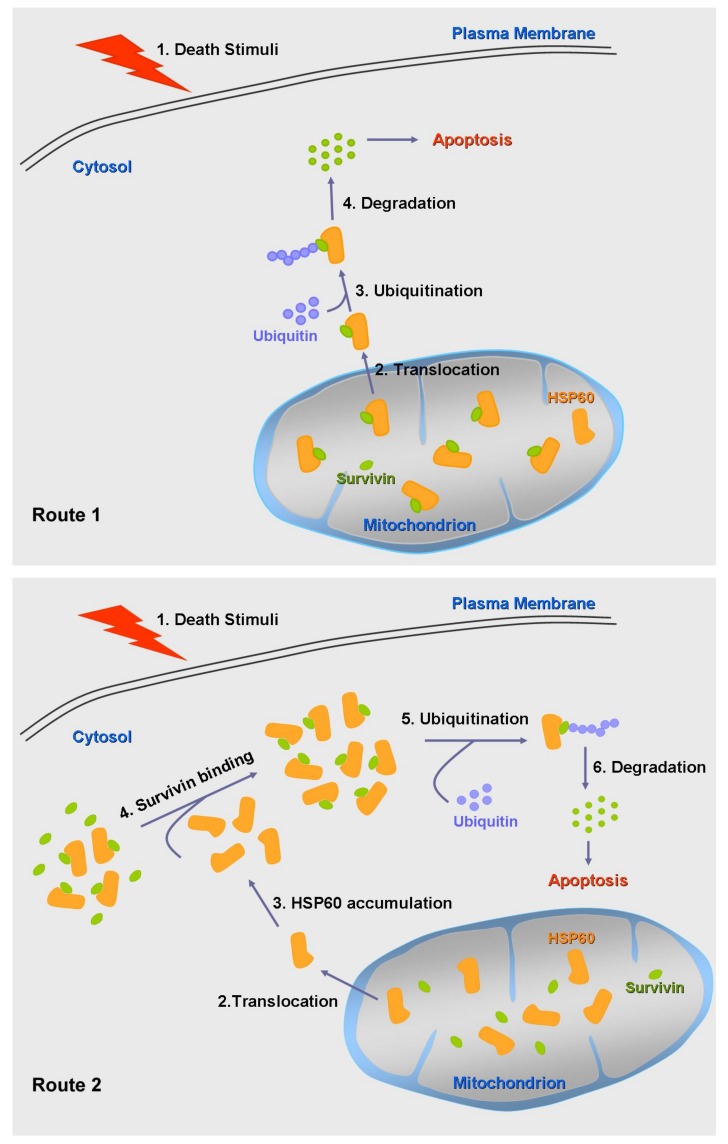
A hypothetical model depicting the routes of HSP60–survivin complexes during apoptosis of tumor cells. Route 1 (**upper panel**), after cell death stimuli, mitochondrial HSP60–survivin complexes translocate to the cytosol facilitating the ubiquitination and degradation of survivin and triggering apoptosis. Route 2 (**lower panel**), after cell death stimuli, mitochondrial HSP60 translocates to the cytosol. Accumulation of HSP60 in the cytosol leads to its capture of free survivin for further ubiquitination and degradation, leading to cell apoptosis. Note: To focus on the mitochondrial HSP60–survivin complex in cells, free molecules of HSP60 and survivin in the cytosol and mitochondrion are not depicted in the upper panel.

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
