# Peer review of "Functional Compartmentalization of HSP60-Survivin Interaction between Mitochondria and Cytosol in Cancer Cells"

_cells, 2019, doi:10.3390/cells9010023_

Round 1
Reviewer 1 Report
The manuscript by Ya-Hui Huang, Chau-Ting Yeh entitled "Functional compartmentalization of HSP60-survivin interaction between mitochondria and cytosol in cancer cells" is a concise summary of the current state of knowledge about the importance of the interaction between HSP60 and survivin as a trigger affecting apoptosis in cancer cells.
The authors attempt to summarize the significance of the interaction of these proteins based on sometimes divergent information on their functioning. They propose two similar but incomplete models of action leading to apoptosis of the tumor cells. At the same time, they point to the need for further research to shed more light on this phenomenon.
According to the reviewer's knowledge, there are no current review articles strictly and extensively devoted to this topic. Thus the manuscript is interesting and worth publishing. However, some minor issues require clarification/refinement before its publication. These issues are listed below.
Lack of consistency in the use of abbreviations, for example:
Line 182-183
(…) MCF-7 (breast cancer cells, expressing cytosolic and mitochondrial survivin) or INS-182 1 cells (insulinoma cells, only expressing cytosolic survivin) result in (…)
The abbreviation is used first and the full name of the cell lines is in brackets, while in other parts of the text the opposite is used i.e. the full name is given first, and the abbreviation is given in brackets.
Line 263
Recently, CCAR2 (cell cycle and apoptosis regulator 2) (…)
The abbreviation is used first and the full name of the protein is in brackets, while in other parts of the text the opposite is used, i.e. the full name is given first, and the abbreviation is given in brackets.
Moreover, the abbreviations are repeatedly introduced into the text, e.g. MCF7.
Line 300
(…) cytosolic survivin-XIAP complex and thus XIAP degradation [56], and (…)
The citation which the authors refer to in this part of the manuscript seems to be improper. Shouldn't it rather be an article by Dohi et al., 2007?
Figure 1.
Since only one mitochondrion is represented in the figure, the singular term (mitochondrion instead mitochondria) should be used.
The upper panel of figure 1. lacks the names of depicted proteins (HSP60, survivin, and ubiquitin).
The unlabeled arrow on the bottom panel should contain the signature e.g. “survivin binding”.
If the proposed models are to be of a general nature and refer to various cancer cells, they must be improved.
The upper panel of the figure does not include the dual subcellular localization of survivin and HSP60. As follows from the text, both proteins are present in the cytoplasm and mitochondria of cancer cells, while in figure 1. only the mitochondrial pools of proteins are included. Exemplary, in MCF-7 cells binding of HSP60 to survivin occurs in the mitochondria but not in the cytosol (line 244). However, these cells express cytosolic and mitochondrial survivin (line 182). What happens to cytosolic pools of these proteins after death stimuli? If the omission of the dual subcellular localization of mentioned proteins is intentional to facilitate focusing on the pools of proteins involved in the pro-apoptotic response to death stimuli, then it should be indicated in the figure description.
Author Response
Dear Reviewer,
We appreciate very much your precious comments. Here are the point-to-point responses.
Lack of consistency in the use of abbreviations, for example:
Line 182-183
(…) MCF-7 (breast cancer cells, expressing cytosolic and mitochondrial survivin) or INS-182 1 cells (insulinoma cells, only expressing cytosolic survivin) result in (…)
Ans: It has been corrected as "... breast cancer cells expressing cytosolic and mitochondrial survivin (MCF-7 cells) or insulinoma cells expressing only cytosolic survivin (INS-1 cells) results in ....”
The abbreviation is used first and the full name of the cell lines is in brackets, while in other parts of the text the opposite is used i.e. the full name is given first, and the abbreviation is given in brackets.
Ans: We have corrected the way of abbreviation usage. The abbreviation was placed in brackets, and the full name was given first.
Line 263
Recently, CCAR2 (cell cycle and apoptosis regulator 2) (…)
The abbreviation is used first and the full name of the protein is in brackets, while in other parts of the text the opposite is used, i.e. the full name is given first, and the abbreviation is given in brackets.
Ans: It has been corrected as " Recently, cell cycle and apoptosis regulator 2 (CCAR2) has ...".
Moreover, the abbreviations are repeatedly introduced into the text, e.g. MCF7.
Ans: The repeatedly introduction of MCF-7 cells has been deleted.
Line 300
(…) cytosolic survivin-XIAP complex and thus XIAP degradation [56], and (…)
The citation which the authors refer to in this part of the manuscript seems to be improper. Shouldn't it rather be an article by Dohi et al., 2007?
Ans: Thank you for finding this mistake. We have corrected it.
Figure 1.
Since only one mitochondrion is represented in the figure, the singular term (mitochondrion instead mitochondria) should be used.
Ans: It has been corrected.
The upper panel of figure 1. lacks the names of depicted proteins (HSP60, survivin, and ubiquitin).
Ans: The individual names have been given next to the depicted proteins in figure 1.
The unlabeled arrow on the bottom panel should contain the signature e.g. “survivin binding”.
Ans: Thank you for the comment. We have already added “survivin binding” as step 4.
If the proposed models are to be of a general nature and refer to various cancer cells, they must be improved.
The upper panel of the figure does not include the dual subcellular localization of survivin and HSP60. As follows from the text, both proteins are present in the cytoplasm and mitochondria of cancer cells, while in figure 1. only the mitochondrial pools of proteins are included. Exemplary, in MCF-7 cells binding of HSP60 to survivin occurs in the mitochondria but not in the cytosol (line 244). However, these cells express cytosolic and mitochondrial survivin (line 182). What happens to cytosolic pools of these proteins after death stimuli? If the omission of the dual subcellular localization of mentioned proteins is intentional to facilitate focusing on the pools of proteins involved in the pro-apoptotic response to death stimuli, then it should be indicated in the figure description.
Ans: In the revised version of figure 1, additional description for the figure are added to the end of legend. That is " Note: To focus on the mitochondrial HSP60-survivin complex in cells, free molecules of HSP60 and survivin in the cytosol and mitochondrion were not depicted in the upper panel."
Reviewer 2 Report
In this manuscript, the authors have reviewed the latest advances on the molecular functions of HSP60 and surviving, and a hypothetical model is proposed on how HSP60-survivin heterocomplexes participate in cell apoptosis upon death stimuli.
The article is well written and the information provided is of broad interest. The overall article could be useful for readers. My only concern refers to the final proposal of the article: “We speculate that survivin will undergo degradation possibly through ubiquitination since the cells have proceeded to apoptosis…The hypothesis remains to be tested and verified in the future.” Actually, there are several publications where the association of both proteins has been shown to stabilize the heterocomplex, and how its disruption promotes protein degradation via ubiquitination (among several other studies, see RJ Kelly et al, Mol Cancer 2011, and more recently W.Kim et al, Int J Mol Sci 2019, just to provide some examples). Perhaps the authors should modify this overstated hypothesis and also include the existing literature in this matter.
Author Response
Dear Reviewer,
We thank you for the positive comments. The manuscript has been modified to address your comment:
Ans: The authors appreciate very much for the suggestions. To address this comment, we added the following sentences to lines 336-341 and cited both papers. “The study by Kim et al., has demonstrated the formation of CCAR2-HSP60-survivin complex, while no detailed experiments were performed to explore the fates of the complex upon death stimuli. [29] A review by Kelly et al suggested chaperon binding to survivin could form stabilized complex. [62] Therefore, it remains possible that the heterocomplex could stabilize both proteins, whereas upon death stimuli, other factors might join in to destabilize them, leading to survivin degradation via ubiquitination.”
Reviewer 3 Report
Cell-664149
Comments
In this article Huang and Yeh have reviewed the role of function HSP60 compartmentalization and their interaction between mitochondrial and cytosol in cancer cells. This review nicely documented several roles of HSP60 and survivin which are over expressed in many cancer cells and inhibit cell proliferation and apoptosis. This review is a compressive for researcher and nicely presented, however it need some revision before it accepted in “Cell”.
Major comments
If author can provide interaction of HSP60 and surviving during apoptosis by a schematic diagram, it would better to understand for reader. Did author review about the role of HSP90 or other HSPs with interaction with surviving in cancer cell proliferation and apoptosis. What are the interaction between HSP60 with proto-oncoprotein and tumor suppressor proteins. HSP60 interacts with many other proteins for cell death and proliferation. Why authors only concentrated the interaction between Surviving and HSP60, do they have experimental evidence from their own work. What about the interaction of HSP70 and surviving? Ref 46 is not much recent, do author has very recent ref regarding this. What is CCAR2?Minor comments:
Please check typos and grammatical errors one more Abbreviation used, please define Author said, “this research has no funding” instead it would be better to say “this review work has no funding”.
Author Response
Dear Reviewer,
Thank you for your comments. We appreciate very much for the suggestions. The point-to-point responses are as the followings.
Major comments
If author can provide interaction of HSP60 and surviving during apoptosis by a schematic diagram, it would better to understand for reader.
Ans: We have presented the hypothesis on HSP60-survivin interaction upon death stimuli in figure 1 in the original submission. The figure has been modified in the revision. If the reviewer still cannot find the figure in the upload, please do contact the editor.
Did author review about the role of HSP90 or other HSPs with interaction with surviving in cancer cell proliferation and apoptosis. What are the interaction between HSP60 with proto-oncoprotein and tumor suppressor proteins. HSP60 interacts with many other proteins for cell death and proliferation. Why authors only concentrated the interaction between Surviving and HSP60, do they have experimental evidence from their own work.
Ans: Because this review article is intended to focus on HSP60-survivin interaction, to avoid confusions, we did not discuss other HSPs. We wrote up this review article because in our previous study on HSP60 and survivin in HCC cells (ref 12), it was found that HSP60-survivin interaction occurred only in the cytosol, different from what had been reported in other cancers (where the interaction occurred in the mitochondria). During literature research, we discovered that there was scanty information regarding the functions/fates of HSP60-survivin complex in cells. Hence, we consider this a worthy project to pursue in the future and there is an ongoing project related to this aspect in our research center.
What about the interaction of HSP70 and surviving?
Ans: There is scanty studies focusing on HSP70 and survivin. In 2008, a study indicated that expression of HSP70 was not correlated with survivin in retinal tissues by using immunohistochemistry (Histol Histopathol 2008; 23:827-31). In 2016, a study showed heterogeneous expression of HSP70/90 and survivin with a significant association between them in retinoblastoma, leaving the question of which (HSP70 or 90?) was associated with survivin (Chem Biol Interact 2016; 252:141-9). Regarding the physical interaction between HSP70 and survivin, only a study in 2011 found that HSP70-survivin interaction occurred in exosome (Apoptosis 2011; 16:1-12). We think this topic is not in the scope of this review.
Ref 46 is not much recent, do author has very recent ref regarding this.
Ans: Ref 46 (An inhibitor of the Interaction of Survivin with Smac in Mitochondria Promotes Apoptosis. Chem Asian J 2019 Nov 18;14:4035-41.) is the latest study on mitochondrial survivin promoting apoptosis.
What is CCAR2?
Ans: The full name and description of CCAR2 was given in section 4.1: Interaction between HSP60 and survivin in mitochondria.
Minor comments:
Please check typos and grammatical errors one more Abbreviation used, please define Author said, “this research has no funding” instead it would be better to say “this review work has no funding”.
Ans: Thank you for the suggestions. The abbreviation and the contents of funding have been corrected.
Round 2
Reviewer 3 Report
The author have provided the sufficient information for my review comments.